# Fabrication and Mechanical Performance of Non-Crimp Unidirectional Jute-Yarn Preform-Based Composites

**DOI:** 10.3390/molecules26216664

**Published:** 2021-11-03

**Authors:** Yeasin Ali, Atik Faisal, Abu Saifullah, Hom N. Dhakal, Shah Alimuzzaman, Forkan Sarker

**Affiliations:** 1Department of Textile Engineering, Dhaka University of Engineering &Technology, Gazipur 1700, Bangladesh; yeasinalitex@gmail.com; 2Department of Yarn Engineering, Bangladesh University of Textiles, Dhaka 1208, Bangladesh; atiknabil@gmail.com; 3Advanced Polymers and Composites Research Group, School of Mechanical and Design Engineering, University of Portsmouth, Portsmouth PO1 3DJ, UK; abu.saifullah@port.ac.uk (A.S.); hom.dhakal@port.ac.uk (H.N.D.); 4Department of Fabric Engineering, Bangladesh University of Textiles, Dhaka 1208, Bangladesh; sazaman_2006@yahoo.com

**Keywords:** natural fibre, UD preform, non-crimp preform, mechanical testing, structural composites, tensile properties

## Abstract

This work developed novel jute-yarn, non-crimp, unidirectional (UD) preforms and their composites, with three different types of warp jute yarns of varying linear densities and twists in the dry UD preforms, in order to present a possible solution to the detrimental effects of higher yarn twists and crimp at the warp–weft yarn interlacements of traditional, woven, preform-based composites on their mechanical properties. In the developed UD preforms, warp jute yarns were placed in parallel by using a wooden picture-frame pin board, with the minimal number of glass weft yarns to avoid crimp at the warp–weft yarns interlacements, which can significantly enhance the load-bearing ability of UD composites compared to traditional, woven, preform composites. It was found that an optimal combination of jute warp yarn linear densities and twists in the UD preforms is important to achieve the best possible mechanical properties of newly developed UD composites, because it encourages a proper polymer-matrix impregnation on jute fibres, leading to excellent fibre–matrix interface bonding. Composites made from the 25 lb/spindle jute warp yarn linear density (UD25) exhibited higher tensile and flexural properties than other UD composites (UD20, UD30). All the UD composites showed a much better performance compared to the traditional woven preform composites (W20), which were obviously related to the higher crimp and yarn interlacements, less load-carrying capacity, and poor fiber–matrix interfaces of W20 composites. UD25 composites exhibited a significant enhancement in tensile modulus by ~232% and strength by ~146%; flexural modulus by 138.5% and strength by 145% compared to W20 composites. This reveals that newly developed, non-crimp, UD preform composites can effectively replace the traditional woven composites in lightweight, load-bearing, complex-shaped composite applications, and hence, this warrants further investigations of the developed composites, especially on long-term and dynamic-loading mechanical characterizations.

## 1. Introduction

The use of natural fibres in various structural and nonstructural composite applications is becoming a compulsory alternative to their synthetic counterparts because of natural fibres’ biodegradability and enormous environmental benefits. Additionally, natural fibres possess some distinctive properties, for example, a low specific weight, good mechanical properties, excellent thermal and acoustic insulation abilities, low production costs, etc., which are advantageous and comparable to conventional fossil-fuel-based synthetic reinforcing materials such as glass fibres, etc. [1,2,3,4,5,6]. Among all the natural fibres used as composite reinforcement, jute is one of the most abundant natural fibres after cotton, which is mainly grown in Bangladesh and other countries, such as India and China [1]. In the last few decades, jute yarns (hessian yarns, etc.) have been produced from locally and nationally sourced raw jute fibres in Bangladeshi spinning industries, which are mainly used to manufacture sacks for agricultural purposes, ropes, upholstery, carpet and many other technical applications. Recently, fibre-reinforced composite manufacturers have been trying to use natural dry-fibre preforms in structural composite applications, wherein the jute yarns can be utilized in a better way and can significantly increase the mechanical performance of the composites.

Traditional textile architectures—woven, knitted, or braided preform structures are used in composites [7,8,9], and it was found that the optimum mechanical strength could not be achieved in these preform-based composites due to the crimp, which is normally generated at the interlacement of warp/weft yarns during the fabrication processes of textile preforms. Therefore, designing the fibre placement technique to remove any crimp is becoming a major concern for achieving optimal mechanical performance in composites. There are plenty of works reported in the literature which are related to the development of natural based continuous or unidirectional dry-fibre preforms for avoiding any yarn interlacement or crimps for composite application [10,11,12,13,14,15,16,17,18,19]. Khondker et al. [20] used a pinboard technique to produce a yarn-based, unidirectional (UD) fibre sheet. These preforms were further impregnated and compressed in order to manufacture composites, but these preforms were only used in manufacturing premoulded composites. However, this technique could not be used for making complex- design-shaped composites. Shah et al. [18] recently developed a continuous yarn-based mat by using a drum-winding technique for fibre collection, and used a cellulose-based binder to hold the fibres in a parallel direction. This preform may tear off due to the low binding force between the cellulose binder and the jute fibre. Weyenberg et al. [15] also used a pinboard to make UD flax yarn composites, where they manufactured the impregnated UD sheet by using a prepregging system. Recently, many studies can be found which are associated with the development of UD fibre-based preforms of both flax and jute fibres [12,13,17,21]. Special attention is needed during the manufacturing of composites from these UD fibre sheets, which increases the cost of these preforms and limits their use to only high-performance applications. The major drawback of the UD preforms of fibres and yarns is the lack of draping ability, and as a result of this, they can be torn off very easily in the presence of a little shear force during manufacturing of 3D composite structures based on these preforms.

Considering the above-mentioned disadvantages of UD preforms and the cost-effective structural composite applications, the design of yarn-based woven preform composites is still in high demand, although these preforms are associated with two major concerns—twist and crimp. Yarn twist is an essential element in the yarn spinning process and can be adjusted based on the yarn count, strength and requirements of the users, whereas crimp of the yarn is formed at the interlacement of warp and weft yarns during the fabrication of woven structures, responsible for the transverse distribution of loading and stress decay for straightening-out of the yarns. A non-crimp, UD preform is desirable for achieving optimum mechanical properties of yarn-based jute-fibre composites. Moreover, inserting weft yarn may result in the formation of crimp in the preform. To reduce the crimp percentage, Vanleeuw et al. [21] recommended thin yarn in the weft direction to produce UD woven fabric, though generation of crimp in the composites was still observed after the consolidation of composites.

In order to utilize natural fibre-based yarns in the structural composite industries, yarns have to be printed in the parallel direction so that a minimal amount of stress concentration and crimp are generated during the loading. To overcome these challenges, constructing a new UD dominated weave architecture with a minimum weft yarn insertion, in order to keep a very low number of warp/weft yarn interlacements and simultaneously assist in holding the original weave shape by resisting shear forces during complex-shaped-composite manufacturing processes, is desirable. Moreover, weft yarn diameter and its linear density are also other baselines that can significantly contribute to composite mechanical properties. To avoid this issue, traditional E-glass filament bundles can be inserted in the weft direction to construct the woven structure of UD jute yarns. In such structures, the use of flat, configured glass filaments in the weft direction does not interrupt the UD orientation of the jute yarns too much and facilitates enough rigidness of the weave structure to hold any desirable shape. Considering this concept, a new idea for manufacturing the jute yarn, non-crimped, UD preform was investigated and analysed in this research work so that a scalable, dry-fibre preform can be produced which is suitable for making any kind of complicated shape without any compromise of the composite’s mechanical performances.

This aim of this study was to develop yarn-based, non-crimped, UD preforms from jute yarns, which facilitate making complicated-shaped composites with the highest possible mechanical performances. Three types of non-crimped, UD preforms were fabricated, following a plain (1/1) weave structure with varying linear densities (yarn count) and twists of jute warp yarns, and using the same type of multifilament glass weft yarns, keeping a very long glass weft yarn spacing (20 mm) for achieving a minimal number of warp/weft yarn interlacements and crimps in the preform structures. Another type of traditional, plain, woven preform structure was also developed, wherein jute warp yarns and glass weft yarns were used with a closer weft yarn spacing (1.5 mm) to create a traditional, more tightly woven structure with a higher number of warp/weft yarn interlacements and crimps. Composites were manufactured using the developed jute-yarn UD and woven preforms; their physical, tensile, flexural and thermomechanical properties were evaluated and compared in order to analyse the mechanical performances of newly developed, non-crimped, jute yarn, UD composites and to understand how these non-crimped, UD composites exhibit significantly higher mechanical properties compared to traditional, plain-woven-structure preform composites.

## 2. Materials and Methods

### Materials

Three types of jute yarns of different linear densities and twists per meter (TPM) were collected from Bangladesh jute research institute (BJRI), Dhaka, Bangladesh, which were spun with 100% Bangla Tossa-C (B.T.C) jute fibres (see Appendix A). Polyester resin (Norsodyne-3351AW) was used as a matrix and equivalent hardener (Methyl Ethyl Ketone Peroxide, Type-A) was used in the manufacturing of the composites. The physical and mechanical properties of the polyester matrix used are presented in Table 1. The collected jute yarns were used as warp yarns in the woven and UD preform structures, while for the weft yarns in both types of preforms, multifilament E-glass fibre was used, which was collected from a local supplier.

## 3. Methods

### 3.1. The Preform Development

Unidirectional (UD), non-crimped preforms were developed using a 450 × 450 mm^2^ hand-crafted wooden frame (see Figure 1). The two edges of the frame contained uniformly arranged metal pins to support the preform warp yarns’ arrangement during the drawing of yarns in the frame board.

In Figure 1, the preform preparation technique is illustrated. At the first stage, the jute warp yarns were placed manually in a unidirectional way (zero-degree direction) in the wooden frame and were locked into the metal pins with adequate tension for at least four hours to observe the stability of the drawn yarns in the frame. Once the UD yarns became stable in the wooden frame without any crimp, the weft glass yarns were inserted using a needle at a 20 mm weft-yarn spacing, following a warp–weft yarn-interlacement sequence of 1/1 plain-woven structure. The preforms were cut and separated after a three-hour relaxation period. Thus, unique, unidirectional (UD), non-crimped, jute-yarn-dominated, dry-fibre preforms were prepared in this study. The area densities and thicknesses of the three types of non-crimped UD preforms produced (UD20, UD25, UD30) are provided in Table 2, which are varied due to the differences in linear densities and TPM of warp jute yarns used in the developed preforms. The other type of jute warp yarn/glass weft yarn preform, with a traditional plain-woven structure (W20, see Table 2) was also developed using the same process, but with a much closer weft yarn-insertion spacing distance of 1.5 mm, which created more warp–weft yarn interlacements, higher crimp and a more tightly woven preform structure.

### 3.2. Composite Fabrication

Figure 2a demonstrates the compression moulding technique. At the start of the composite manufacturing, the two layers of preforms were placed separately between the two metal plates of the pressing machine for five minutes at room temperature, with a pressure of 14 MPa, in order to undertake calendaring action on the preforms so that the yarns in the preforms would be flat, and hence remove any crimp, especially in jute yarns. The polyester resin was prepared with a resin-to-hardener ratio of 100:1.5 and was applied manually on both sides of the calendared preforms by a wet-laid method. Following this, the two wet preforms were arranged into the pressing machine on top of each other. Finally, a 14 MPa compression pressure was applied at a temperature of 100 °C for 30 minutes. This process ejected out the extra resin from the compressed jute layers, cured the remaining resin and consolidated the composites. After that, the hardened composite was removed from the pressing machine and left 24 hours at room temperature to complete the curing process further. All composites were made following the same procedures, and the details are provided in Table 3. They are coded according to the codes of the preforms used in their manufacturing process.

### 3.3. Measurement of Fibre Volume Fraction and Composite Density

Composite densities were measured with an AJ5OL analytical balance (Mettler Toledo, Leicester, UK), following the ASTM-D3800-99 standard. The water immersion method was used for measuring the densities of the composites and the edges of the measured composite samples were covered carefully with the epoxy resin to inhibit water penetration into the samples. Following this method, the samples were weighed in the air before being submerged in water individually. To obtain the experimental densities of the composites, Equation (1) was used.
ρ_C_ = (ρ_f_ × ρ_m_)/(ρ_m_ × W_f_ + ρ_f_ × W_m_)(1)
where, ρ_c_ is the measured density of the composites, ρ_f_ is the density of the fibre, ρ_m_ is the density of the matrix, W_f_ is the weight fraction of the fibre and W_m_ is the matrix fraction in composite.

The fibre volume fractions of the composite were calculated after finding the density of the composites. The resin weight of a composite can be calculated by subtracting the known weight of the preform from the composite weight, and hence the fibre volume fraction (V_f_) can be obtained using the following Equation (2).
V_f_ = ρ_C_ × (W_f_/ρ_f_)(2)

### 3.4. Characterisation of Composites

#### 3.4.1. Tensile Test

The tensile tests of composite specimens with a 150 mm length, 15 mm width and 1.5 mm thickness were performed according to the ASTM D3039 standard, using a 5 mm/min displacement rate and a properly calibrated 10 kN load cell in the AG-X plus Japan universal strength-testing machine. For each composite type, five specimens were tested. Figure 3 depicts the loading of tensile specimens in the universal testing machine.

#### 3.4.2. Flexural Test

ASTM-D790 standard was used to perform the three-point-bending flexural tests of composite specimens with a 172 mm × 12.4 mm × 1.5 mm dimension, using the universal testing machine. The bending support length (load-span) and the deflection rate were 96.3 mm and 1.4 mm/min, respectively, for the flexural tests. The flexural test setup can be seen in Figure 4.

#### 3.4.3. Dynamic Mechanical Thermal Analysis

A double cantilever bending mode DMA (TA Q-800 instrument) machine was used to determine thermomechanical properties of the composites at 1 Hz oscillation frequency in nitrogen atmosphere. The machine was run with a temperature range from 25 °C to 180 °C at a constant temperature ramp of 2 °C/min. From these DMTA experiments, the storage modulus (E′) and loss factor (tan δ) of the tested specimens were determined.

#### 3.4.4. Scanning Electron Microscope

Composites’ morphologies and fracture surfaces were examined with gold-coated specimens using scanning electron microscopy (ZEISS). The operating voltage of the machine was 12 kV, as recommended for natural fibres in the literature [22].

## 4. Results and Discussion

### 4.1. Physical Properties of Yarns, Developed Preforms and Composites

Table 2 provides experimentally obtained physical properties of jute warp yarns, glass weft yarns and the developed preforms utilizing them. In order to assess the quality of jute warp yarns used in this study, linear densities and TPMs were measured. Both of these values of the jute yarns increased with the increase in yarn counts (lbs/spindle) (see Table 2) and these variations resulted in differences in the thicknesses and area densities of the developed UD20, UD25, UD30 preforms, since the same type of glass weft yarns with a constant 20 mm weft yarn-insertion spacing distance were used in these preforms. Between them, UD20 preform displayed the lowest values of thickness and area density, which was due to the compactness of jute yarns, after imparting higher amount of twist in the yarns, whereas with the increase in linear densities, the amount of imparted twists in the yarns were reduced which ultimately increased the loose appearance of the fibres in the respective jute yarns used in UD25 and UD30 preforms. The W20 preform showed a relatively higher thickness and area density compared to other preforms, which was due to the use of a much closer glass weft yarn spacing (1.5 mm) for the higher interlacement of warp and weft yarns in its plain-woven structure and also a higher density of glass fibres, resulting in a much more tightly woven structure compared to UD20, UD25, UD30 preforms. Thus, the measured differences of physical properties of the developed preforms were expected to correlate with the performance of their composites developed in this work.

Table 3 provides the physical properties of manufactured composites. Considering the fibre volume fraction, it was observed that W20 composites had the lowest value, at 38% only. Several reasons could be identified as the contributing factors to this woven preform composite having the lowest volume fraction. Firstly, the balance and tighter architecture (warp × weft yarns) inhibited the proper flash out of resin due to the high amount of crimp formation in the preform. Secondly, twist in the jute warp yarn used in the preform was also the highest (see Table 2), which did not allow the polyester matrix to reach the core of the fibre, rather, matrix impregnation occurred, and the matrix was trapped around the yarns. Thus, crimp and twist increased the matrix-rich area in the W20 composite. On the contrary, all three UD preform composites showed a higher amount of fibre volume fractions over the W20 composites, which might be due to the fewer interlacements of warp/weft yarns, loose weave structure and lack of crimp in their preform structures. It was also found that fibre volume fractions were increased with the linear densities of yarn used in UD composites due to the lowering of imparting twist in the respective yarns. All of the composites showed a small percentage of voids in the composites structure, which could be related to the lower viscosity of the polyester resin matrix used in the composite manufacturing process [23].

### 4.2. Tensile Properties

Tensile behaviour of a fibre-reinforced composite is considered the most desirable feature to use to evaluate the performance requirements when selecting a suitable engineered application. Generally, the tensile properties of natural fibre-reinforced composites are exceptionally dependent on a number of fibre properties such as fibre length and orientation, fibre–matrix interfacial bonding, aspect ratio, moisture absorption tendency, morphology, and dimensional stability [24]. Additionally, the physical aspect of fibres and tensile properties of individual fibres have a considerable impact in determining the ultimate strength of the composites [23]. Results obtained from the tensile tests are presented in Appendix A and Figure 5.

Though single jute-fibre tensile properties have a large scattering effect due to irregularity of fibre fineness at the composite scale for each specimen type, we did not observe any such significant variation in the tensile properties. The variation in tensile properties are not statistically significant (*p* > 0.05). While conducting statistical analysis, a low percentage of coefficient of variation (less than 10%) of the properties was observed.

The typical stress–strain curve clearly confirmed that composites made from non-crimped UD preforms consistently showed brittle failure, in a way similar to other composites made from reinforcing natural fibres reported in the literature [18,23]. From the typical stress–strain curve it can be seen that stress developed progressively in the elastic zone and then created a pseudo-ductile zone before reaching maximum stress and failing suddenly for all composite types. Similar behaviour in natural fibre composites made from yarn architecture was also observed in the previous study [1]. The failed samples in Figure 5d clearly show the uniform failures in tensile testing.

Comparisons of tensile strength and modulus of different composites are graphically presented in Figure 5b,c. From the figure, it can be observed that UD20 composites showed relatively lower tensile properties (tensile strength 94 MPa and modulus 5.6 GPa) compared with other UD composites—UD25 and UD30. UD25 composites exhibited the highest tensile strength (133 MPa) and tensile modulus (8.3 GPa) among all types of UD and woven (W20) composites, while UD30 composite had almost a similar tensile strength (125 MPa) and modulus (7.8 GPa) to UD25 composites. The inferior tensile performance of low linear-density jute-yarn-reinforced composite UD20 was related to the yarn spinning process and the impregnation of resin on jute yarns’ surfaces in the preforms during the composite manufacturing process. Finer quality jute yarns with low linear density are normally produced with an excessive rate of mechanical actions in the carding section, to ensure thinner jute yarns during the spinning process. In addition to that, a higher amount of twist is also imparted to finer jute yarns to ensure the dimensional stability and load-carrying capacity of the yarns. However, with the increase in twist level, the strength of the yarn decreases owing to the increased angle between jute yarn and loading direction. Ma et al. [2] reported that textile yarns based on hemp and sisal fibres are directly affected by different stages of spinning operations, and tensile properties are significantly reduced with an increase in twist level in single-plied yarn. A higher twist level in yarns incurs more resin-rich area in composites, which may generate stress concentration during the application of load in the tensile direction.

From the comparative analysis of UD20 and UD25 composites, it became obvious that with the increase in linear densities and reduction in TPMs of jute yarns in the UD preforms, fibre volume fractions and tensile strength of the composite were increased. UD25 composite showed an improvement of 41.5% and 47% in the tensile strength and modulus values compared to UD20 composites. The reduction in twist level in the jute yarns of UD25 composite allowed the jute fibres to be arranged slightly more in parallel than the UD20 composites. Additionally, higher linear densities of 25 lbs/spindle jute yarns in UD25 composite ensured a better resin impregnation into the fibres and thus enabled the impregnated fibres to resist the maximum amount of load during tensile testing. However, comparing UD25 and UD30 composites, the increase in tensile properties could remain almost similar or even be slightly reduced with the further increase in yarn linear densities (from 25 lbs/spindle in UD25 composites to 30 lbs/spindle in UD30 composites) or decrease in yarn twist level in the preforms. This drop in tensile properties might be the result of the higher diameter of the jute yarns in higher linear densities and lower twist level in the yarns. The higher diameter resulted in an inferior resin impregnation into yarns during the composite fabrication process, inducing more defects in the composites, as found by the other authors work reported in the literature [2,4]. This reasoning could also be linked to the similarly measured fibre volume fractions of UD25 and UD30 composites. Therefore, an optimal amount of twist level and linear density in the jute yarns is desirable while designing jute-yarn UD composites.

We also compared the tensile properties of our developed non-crimped, UD, jute-fibre preform-based composites (UD20, UD25, UD30) with traditional plain-weave, higher warp–weft yarn interlacements and crimp architecture preform composites (W20). In Figure 5, it can be seen that W20 composites exhibited the worst mechanical performance regarding tensile strength and modulus, with even lower tensile properties than the UD20 composites. This was related to, firstly, the fact that the orientation of yarns in the traditional woven-structured W20 composites formed a wavy appearance due to the higher number of intersections of warp and weft yarns. This resulted in poor tensile properties as the tensile stress was distributed in both warp and weft directions in composites. Secondly, due to the higher interlacement of warp and weft yarns, a higher crimp was also formed in the woven W20 preforms, and upon tensile loading of W20 composites, the yarns were needed to be straightened out before carrying loads, Thus, stress decay occurred in the case of W20 composites.

### 4.3. Flexural Properties

Flexural properties are very important properties for a composite structure because when the flexural load is applied to a component, all three materials’ basic stress states such as tensile, compressive and shear, are induced simultaneously [23,25]. So, composite materials’ performance assessment can be driven strongly by means of flexural-property evaluation.

Moreover, for the composites that are applicable to quantify the structural employment, flexural properties are considered a predominant factor. The flexibility of the materials is determined under static bending conditions following the flexural properties, and greater flexural strength results in brittle and stiffer products [26]. In Figure 6a, a typical flexural stress–strain curve of UD composite specimens is illustrated. Figure 6b,c demonstrates the flexural strength and modulus of the composites made from different jute-warp-yarn quality and preform architectures. All types of non-crimped, jute, UD composites showed very close flexural strength values, although UD30 composites maintained comparatively a slightly higher flexural strength of 178.34 MPa. These UD composites also exhibited the same trend in flexural modulus values as their tensile properties. Combining flexural strength and modulus, UD25 composites (using 25 lbs/spindle jute yarns in preforms) showed a more bending performance than UD20 and UD25 composites, because of the use of optimal twist and linear density in the jute yarns of their preform structure, as described earlier. Comparative analysis between jute-yarn, non-crimped, UD composites and traditional woven (W20) composites revealed that the flexural properties dropped significantly for the latter, in terms of both flexural strength (70 ± 7 MPa) and modulus (2.7 ± 0.25 GPa) (see Appendix A and Figure 6), even less than that of UD20 composite’s flexural strength (172 MPa) and modulus (5.06 GPa) values. The reasons for the poor flexural strength and modulus of the woven composite were due to its traditional woven structure with a higher warp/weft yarn interlacement and crimp, also described in the previous section for tensile properties details.

### 4.4. Theoretical Analysis of the Tensile Properties of Composites

Theoretical analysis of a composite material is often performed to predict the engineered properties of the materials and to compare them with the experimental properties. If the results remain the same, then the design of the experiments is trusted and widely accepted. The following properties have been theoretically studied:

Critical fibre length (l_c_), original fibre length (l_f_), diameter of fibre (d_f_) and aspect ratio (l_f_/d_f_), can greatly influence the mechanical properties of composites. More specifically, these variables help to determine the length efficiency factor, which is mainly responsible for the transfer of stress and stiffness from the fibre to the composite. A composite will carry a maximum amount of load when the critical fibre length is greater than the original length of the fibre (l_c_ > l_f_). Critical fibre length of different yarn quality is determined by using Equation (3), where σ_f_ is the ultimate tensile strength of the fibre, d_f_ is the measured diameter of the fibre of yarn and τ is the interfacial shear strength (IFSS) between the fibre and matrix. The value of IFSS was taken from previously reported work of jute fibres considered untreated [5].

In this study, critical fibre length (l_c_) was found to range from 0.53–0.61 mm, which means it increases within the increase in fibre linear density. This value as reported in the literature ranged from 0.28–0.52 mm [18]. There are some other studies on natural fibre composites where relatively higher critical fibre length l_C_ was observed, which ranged from 0.9–2 mm [27,28,29]. Our study therefore agreed with previous studies on natural fibre composites. Composite researchers who are working on natural fibre composites would like to predict the composite tensile properties such as strength and stiffness by using length efficiency factor ɳ_iS_, and ɳ_iE_, respectively [4,18,30].
l_C_ = (σ_fxD_/2 × τ)(3)
ɳ_IE_ = 1 − {tanh (β l_f_/2)/β l_f_/2}, β l_f_/2 = 2 × (l_f_/D) × √{Gm/E_f_ ln(k/V_f_)}(4)
G = E_m_/2(1 + ϑ_m_)(5)
ɳ_iS_ = 1 − (l_c_/2l_f_)(6)
σ′_m_ = E_m_ε_c_(7)
E_C_ = (ɳ_0_ɳ_IE_ E_f_ V_f_ + E_m_ V_m_)(8)
σ_c_ = (ɳ_0_ ɳ_IE_σ_f_ V_f_ + σ′_m_V_m_)(9)

Cox shear lag model is usually used for calculating the length efficiency factor for stiffness (ɳ_IE_) [31]. It is assumed that the axial loading of fibres and the elastic-stress transfer between the fibre and matrix shows an isostrain behaviour, and by using Equation (4), ɳ_IE_ can be calculated. Here, G_m_ is the shear modulus of the matrix which was calculated from the information provided by the supplier, E_f_ is the measured stiffness of the fibre and k is the constant regarding the maximum packing capacity or fibre volume fraction of the fibre in composites. In this study we assumed that fibres are parallelly arranged in continuous form. Some previous work on natural fibre composites based on yarn architecture also considered the same condition, and for this, the value of constant K used in this study was 0.785 [18,30]. Moreover, the Kelly and Tyson model is used for the calculation of the length efficiency factor for use in calculating strength, where it is found that l_f_ > l_c_ [32]. The calculated length efficiency for the stiffness and strength of different quality of yarns was found to be 0.997 and 0.999. The unity value of the length efficiency factor is related to the higher aspect ratio of fibre and the low critical fibre length. Shah et al. [18] had observed this value to be 0.977 and 0.952 for calculating stiffness and strength, respectively. Madsen et al. [4] calculated the stiffness length efficiency factor at ɳ_IE_ > 0.93 and observed that it changed within the changes of fibre aspect ratio. Moreover, fibre composition, fibrillar networks, crystallinity of fibre, and fibrillar angle may affect the variation of length efficiency factor of yarns. Such an agreement with the previous studies indicates that the fibre in the yarn can ensure higher length efficiency factors in order to secure the load-transfer abilities in composites. In this study of unidirectional composites, we assumed that fibre failure strain is equal to that of composite failure strain by considering isostrain conditions. Thus, failure strain of matrix σ′_m_ can be calculated by using Equation (7). Equations (6) and (7) were used to theoretically calculate the value of stiffness and strength of the composite using a modified rule of mixture [33]. In using these equations, ɳ_0_ is for considering the fibre orientation factor, which is 1 for UD fibre orientation and 0.5 for woven fibre orientation, respectively. The results indicate that the composites’ tensile properties in the experimental data were almost similar to the tensile properties of the composites calculated by using the modified rule of mixture (see Table 4). Differences in the tensile properties of theoretical and calculated results are related to the irregularities present in the fibre, as void content exists in the composite. As a result, material fails in the weakest zone present in the fibre, and thus a lower experimental value is expected compared to theoretical values. From these observations, it can be said that UD composites manufactured from different linear densities of jute yarns are suitable enough to show adequate reinforcing ability and efficient load transfer from fibre to matrix.

### 4.5. Fractographic Observation

Scanning electron microscope (SEM) was used to examine the fracture surfaces of tensile tested specimens, and the SEM images are shown in Figure 7. The intention of the SEM study was to justify the degree of interfacial adhesion between the fibres from different quality jute yarns and polyester matrix. It is clearly visible in Figure 7a that no or a little amount of polyester matrix was present or adhered onto the surface of jute yarn with linear density of 20 lb/spindle in UD20 composites, showing a poor interfacial adhesion between the fibre and polyester matrix in UD20 composites. However, UD25 composites showed an excellent form of fibre failures, where it is clearly visible in Figure 7b that almost all of the fibres were broken linearly. Besides this, a compaction of the fibre inside the yarn was noticed in this type of composite with enough resin impregnation (see Figure 7b). In addition, a large amount of polyester matrix was clinging to the surface of single jute fibres, which confirmed the strong interfacial adhesion of the jute-fibre composite. It was also expected that besides the interfacial adhesion, a strong mechanical interlocking between the fibre and matrix was also formed for UD25 composites. On the other hand, the fracture mode of UD30 composites made from 30 lbs/spindle jute yarn which used preforms displayed a mixture of failure modes (fibre pull out and fibre breakage). From the arrow marks (see Figure 7c), it is clearly visible that not only were a large percentage of single fibres broken uniformly, but also a significant number of single fibres also came out due to the improper impregnation of the matrix on jute fibre. In the case of the W20 composite, we observed excess resin-rich areas due to the higher number of interlacements of warp and weft yarns (see Figure 7d) compared to any of the non-crimped UD composites developed in this work. Moreover, a large amount of fibre pull out was visible on the fracture surface of W20 composites, which clearly indicated the poor interface between the jute fibre and polyester matrix. Strong interfacial adhesion normally allows for a uniform stress transfer from the matrix to the fibre, upon which the tensile properties of composites increase. On the other hand, a poor interface between the fibre and polymer matrix creates stress concentration, which is also responsible for the early failure of composites.

### 4.6. Thermomechanical Analysis of Composites

Based on the previous results, it was decided to carry out thermomechanical analysis on only three types of non-crimped, jute, UD composites, since the traditional woven W20 composite clearly demonstrated very low mechanical performances compared to UD composites. To understand the thermomechanical behaviour at a wide range of temperatures, stiffnesses, damping and interfacial properties of tested composite specimens, dynamic mechanical analysis (DMA) was carried out. Storage modulus and Tan Delta values were measured from DMA tests and are presented in Figure 8 and Figure 9, respectively as a function of temperatures. Storage modulus (E′) is defined as proportional to the stored energy during a cycle in a DMA test, and from this modulus value, the stiffness, degree of crosslinking and fibre–matrix interfacial bonding properties of composites can be described. In the storage modulus figure (see Figure 8), it can be seen that all of the tested composite specimens showed a slow decline with the increase in temperatures at the start of the test, due to the polymer-matrix chains becoming flexible at higher temperatures. After that, the tested composite specimens exhibited a sharp reduction in the storage modulus values in the glass transition region (Tg) of polymer matrix, due to the polymer chain mobility above the Tg temperature. The 20UD composite specimens showed the lowest storage values, while 30UD composite exhibited the highest values, although 25UD and 30UD composite specimens had almost the similar E′ values over the tested temperature range. These results indicate the improved interfacial properties at the fibre–matrix interface of 25UD and 30UD composite specimens, which are related to the differences in yarn diameter and types used in this work, which enable 25UD and 30UD composite specimens to carry and transfer more mechanical stresses. This result also supported the earlier findings in tensile and flexural tests of UD composites, where 25UD composites showed the best performance in both tensile and flexural properties, while UD30 composites exhibited a similar value of these properties to UD25 composites.

The damping factor, Tan Delta (tan δ), is expressed as the ratio of loss modulus (E″, energy dissipated) to storage modulus (E′, energy stored) in the tested composites during a DMA test cycle, and provides information on energy absorption or dissipation mechanisms and glass transition temperature (Tg), etc. The Tg value can be determined from the temperature at which the maximum

Peak of the tan δ curve is observed [34]. For all tested composites, tan δ curves were seen to increase with the temperatures, reach peak values in the glass transition region and then decrease above the glass transition region (rubbery region) (see Figure 9). Higher tan δ values were measured for the 20UD composite specimen compared to 25UD and 30UD composite specimens, as expected, and was related to its higher energy dissipation capabilities. The 25UD composites showed the lowest tan δ values, which was expected because of their higher stiffness, strength in both tensile and flexural testing, and also the better adhesion and bonding of the fibre–matrix interface compared to other UD composites. The 20UD composite specimens also showed a higher Tg temperature (70 °C), whereas the Tg values were identified as 65 °C and 64 °C for 25UD and 30UD composite specimens, respectively. This reduction in Tg values for the 25UD and 30UD composite specimens could be linked to the better fibre–matrix interfacial adhesion in their composites compared to UD20 composites.

## 5. Conclusions

This study successfully developed a novel process for the manufacturing of non-crimped, jute yarn-based, UD preforms, by inserting a small number of glass weft yarns and maintaining a very long weft yarn spacing (20 mm) to keep a minimal warp–weft yarns interlacement and crimp in the preform structure. Three different linear densities of jute warp yarns were used in developing UD preforms, and their effects on the tensile, flexural, and thermomechanical properties of UD composites were studied. The newly developed three types of UD preform composites were also compared with another traditional, high number of jute warp/glass weft yarns-interlacing, woven preform composite. The result indicated that the non-crimped, jute yarn, UD composites displayed a significant improvement in tensile and flexural properties compared to the traditional, high warp/weft yarn-interlacing, woven-architecture jute-yarn composite. This improvement was concluded as the factor with the biggest impact on the lowering of crimp formation in the newly developed UD preforms, which directly enhanced the load-bearing ability of their composites. While comparing three linear densities-based UD jute-yarn composites with the same properties, it was found that UD25 composites made from 25 lbs/spindle-yarn preforms showed the highest tensile, flexural and thermomechanical properties, which means that UD25 composites are the strongest, stiffest composites. The highest properties of UD25 composites were correlated with the optimal amount of twist and linear densities in the jute yarns, leading to an enhancement of fibre volume fraction in the composites, proper matrix impregnation into the yarns, fibre–matrix bonding and the ability of stress transfer between the fibre and matrix. The SEM analysis and DMA results also supported the findings from the tensile and flexural tests. The analytical study also confirmed very similar tensile properties of UD preform-based composites to their experimentally obtained tensile results. Thus, these newly developed, non-crimped, jute, UD, dry preforms can be used as a suitable alternative to traditional woven-architecture composites, creating high load-bearing natural yarn-reinforced sustainable composites and promising alternative material for lightweight structural applications, such as automotive, marine and sporting goods. These newly developed, non-crimp UD composites will be investigated further in the future to better understand their long-term and dynamic mechanical loading (impact, fatigue) behaviours, so that they can be used in the above-mentioned applications with informed durability.

## Figures and Tables

**Figure 1 molecules-26-06664-f001:**
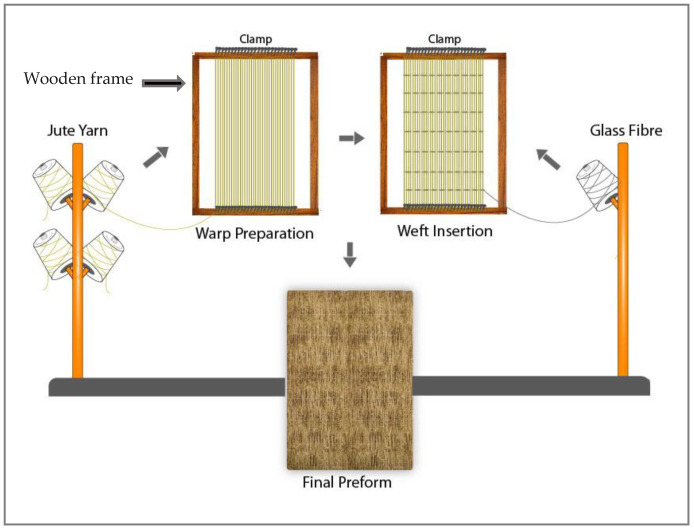
Illustration of the unique preform preparation technique used in this study.

**Figure 2 molecules-26-06664-f002:**
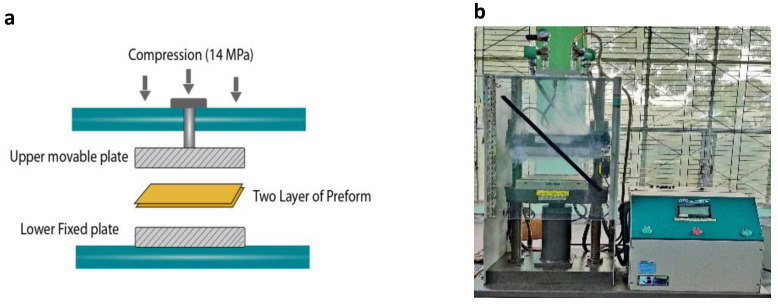
(**a**) Compression moulding schematic process used to fabricate composite laminates (**b**) Carver compression machine used in this study.

**Figure 3 molecules-26-06664-f003:**
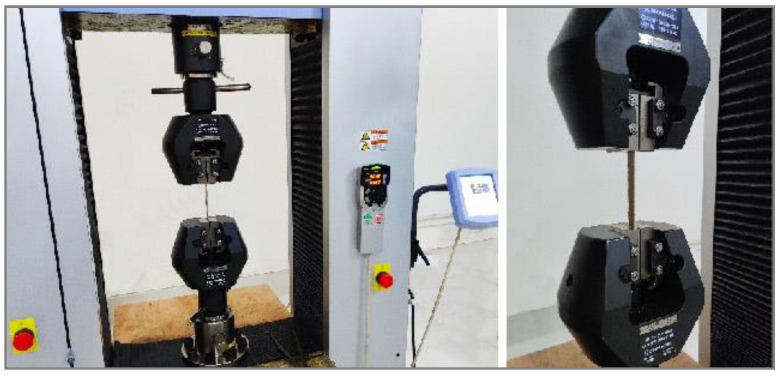
Tensile specimen setup in universal testing machine.

**Figure 4 molecules-26-06664-f004:**
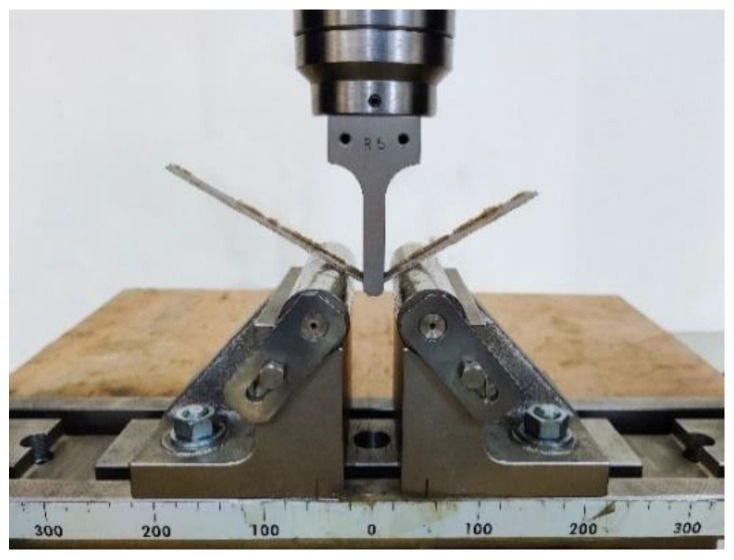
Flexural specimen setup in universal testing machine.

**Figure 5 molecules-26-06664-f005:**
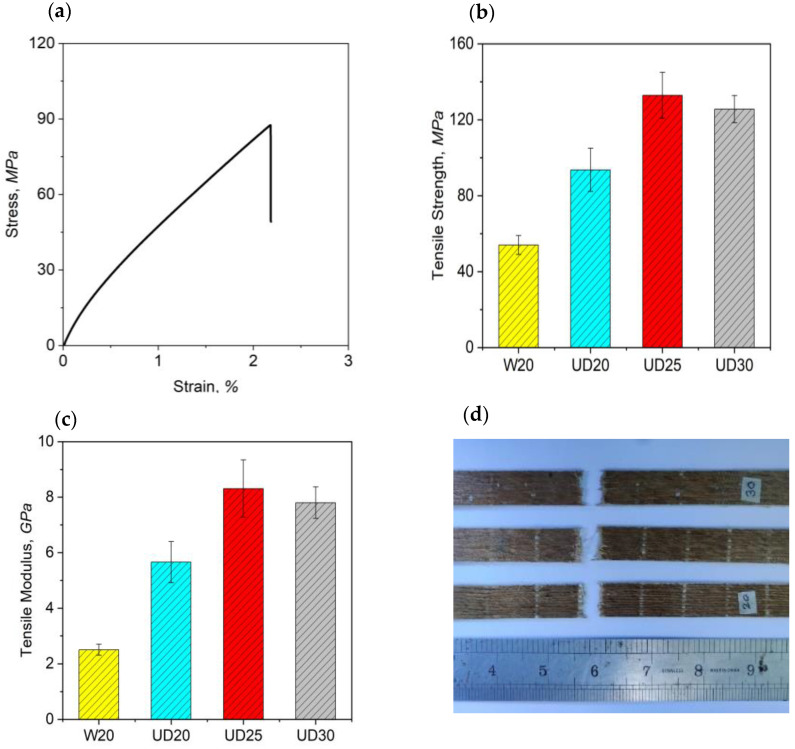
(**a**) Typical stress–strain curve of UD composites; Tensile (**b**) strength and (**c**) modulus of woven (W20) and UD composites (UD20, UD25, UD30); (**d**) UD composites after tensile tests.

**Figure 6 molecules-26-06664-f006:**
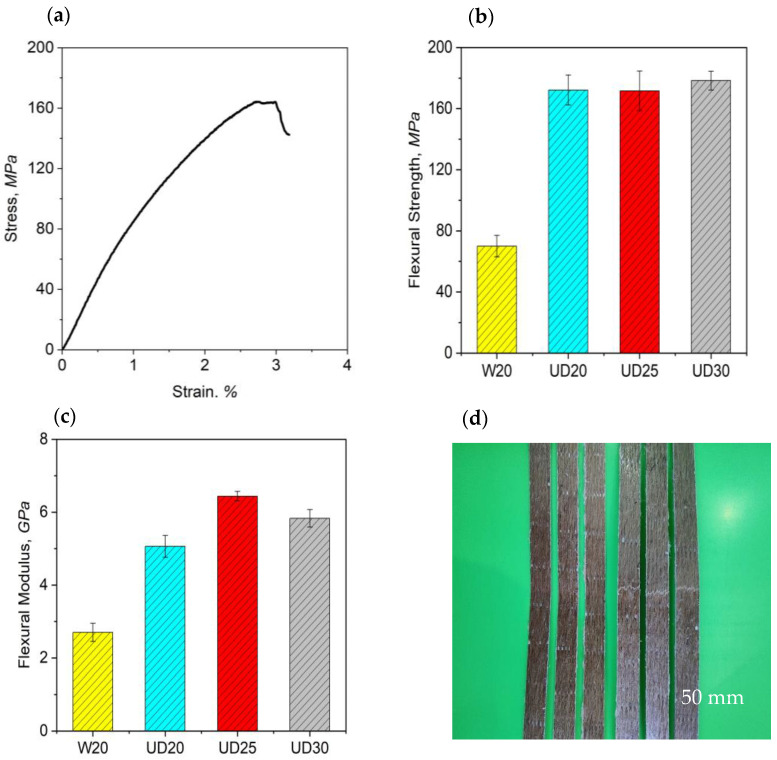
(**a**) Typical flexural stress–strain curve; flexural (**b**) strength and (**c**) modulus of woven and UD composites; (**d**) UD composites after flexural tests.

**Figure 7 molecules-26-06664-f007:**
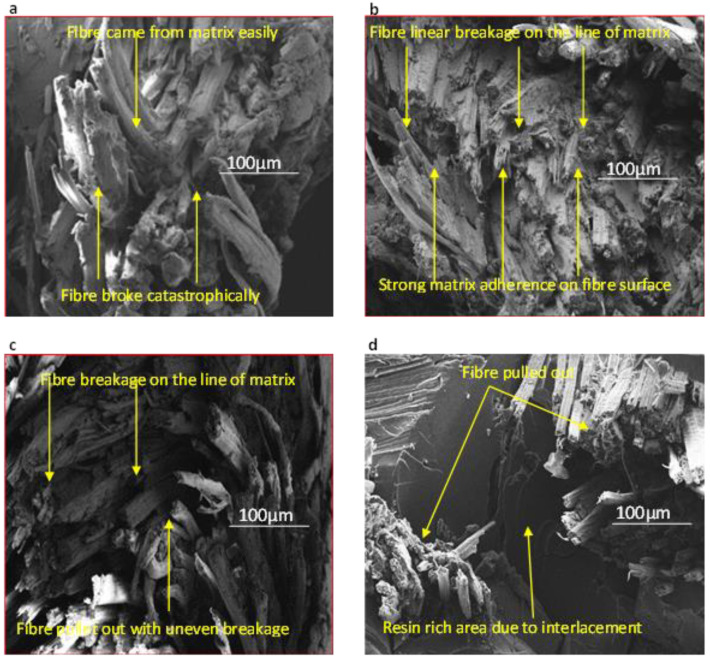
SEM images of fractured specimens after tensile test: (**a**) UD20 composite (**b**) UD25 composite (**c**) UD30 composite (**d**) W20 composite.

**Figure 8 molecules-26-06664-f008:**
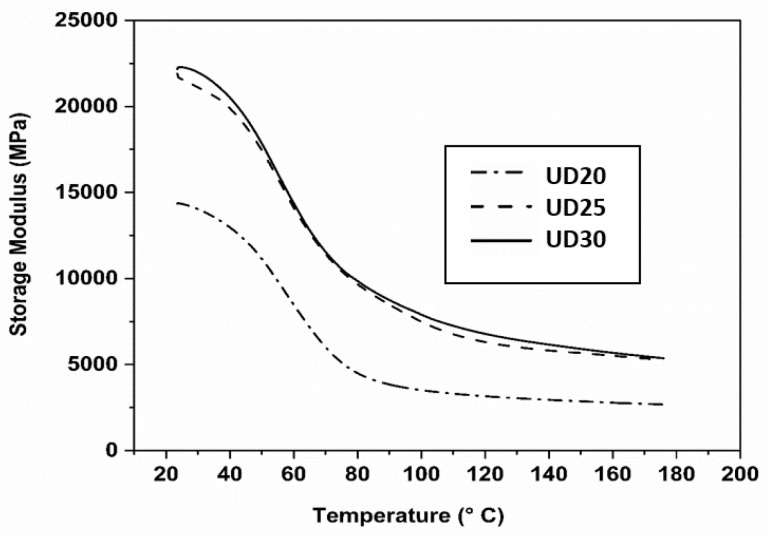
Storage modulus curves of DMA-tested UD composite specimens at a temperature range from 20 °C to 180 °C.

**Figure 9 molecules-26-06664-f009:**
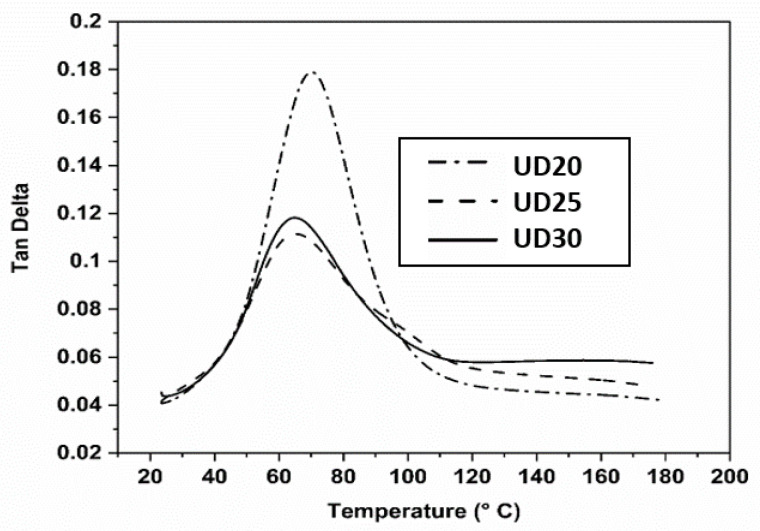
Tan Delta curves of DMA-tested UD composite specimens at a temperature range from 20 °C to 180 °C.

**Table 1 molecules-26-06664-t001:** Physical and mechanical properties of polyester matrix (from supplier datasheets).

Resin	Viscosity [mPas]	Gel Time at 25 °C [mins]	Cured Density ρ_m_ [gcm^−3^]	Tensile Modulus E_m_ [GPa]	Tensile Strength σ_m_ [MPa]	Failure Strain ε_m_ [%]
Polyester	160	85	1.12	3.58	66	2.5
Glass fibre	-	-	2.6	73	3400	2.25

**Table 2 molecules-26-06664-t002:** The physical properties of the developed preforms and the yarns used in this study.

Type of Preform	Warp Jute Yarns			Weft Glass Yarn	Preform Thicknesses [mm]	Preform Area Density [g/m^2^]
	Yarn Direct Count [lb/spyndle]	Linear Density [Tex]	Twist per Meter [TPM]	Weft Yarn Spacing [mm]		
W20	20	701 (±65)	171 (±25)	1.5	2.31 (±0.25)	950 (±102)
UD20	20	701 (±65)	171 (±25)	20	1.39 (±0.15)	540 (±95)
UD25	25	877 (±92)	94 (±14)	20	1.67 (±0.22)	641 (±74)
UD30	30	1052 (±89)	78 (±17)	20	2.31 (±0.26)	854 (±88)

**Table 3 molecules-26-06664-t003:** Measured and calculated physical properties of the composites.

Composite Type	No of Layer	Weight of Fibre [gm]	Composite Weight [gm]	Fibre Volume Fraction [V_f_]	Composite Density Measured [g/cm^3^]	Composite Density Theoretical [g/cm^3^]	Void Volume Fraction [%]
W20	2	23	62	38.45	1.22	1.26	2.80
UD20	2	27.19	51.23	51.37	1.28	1.30	2.14
UD25	2	34.59	57.17	58.86	1.30	1.32	1.85
UD30	2	44.58	72.46	59.88	1.31	1.33	1.95

**Table 4 molecules-26-06664-t004:** Physical, mechanical properties of jute fibre used in this study.

Composite	Matrix Resin	Fibre Tensile Modulus GPa	Fibre Tensile Strength MPa	Fibre Length l_f_	Fibre Diameter D	Fibre Aspect Ratio L_f_/D	Critical Fibre Length l_c_	Length Efficiency for Stiffness ɳ_IE_	Length Efficiency for Strength ɳ_iS_	Theoretical StiffnessE_c_ GPa	Theoretical Tensile Strength σ_c_ MPa
W20	POLYESTER	10	170	220	53	4150	0.53	0.997	0.999	3.9	72
U20	POLYESTER	12	170	220	53	4150	0.53	0.997	0.999	7.2	118
U25	POLYESTER	15	200	216	48	4500	0.63	0.997	0.999	9.4	146
U30	POLYESTER	12	190	170	45	3777	0.61	0.997	0.999	8	131

## Data Availability

Not applicable.

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
