# Peer review of "Fabrication and Mechanical Performance of Non-Crimp Unidirectional Jute-Yarn Preform-Based Composites"

_molecules, 2021, doi:10.3390/molecules26216664_

Round 1
Reviewer 1 Report
This article (Manuscript number molecules-1439462) presents information about the preparation procedure of jute yarn non-crimp unidirectional (UD) preforms and their composites. On the whole, the overall originality of the concept used here is medium-high. Nevertheless, I would recommend publication of this article in Molecules on the condition a major revision of the manuscript will be carried out and the following points will be taken into consideration.
Detailed comments:
1. The abstract needs to be well written with future prospects of the work and describe in short the concept of preparation of yarn non-crimp unidirectional preforms.
2. There is a lack of important references related to the e.g. methods used for receiving presented material. Furthermore, the introduction should be worked out - so as to show the full state of knowledge on this topic.
3. More detailed results discussion should be provided. The discussion section appears to be a collection of data, however, the author’s self-opinion is of importance while drafting a results section of this type.
4. The conclusion reflects an overall summary of the field with further extension and includes future prospective - I would suggest clarifying this section.
5. The style and grammar leave much to be desired in many places, some parts of the text are difficult to understand.
After completing the above-mentioned corrections this work will be more readable. Therefore, it will be useful for the readers of Molecules.
Author Response
Overall Comments: This article (Manuscript number molecules-1439462) presents information about the preparation procedure of jute yarn non-crimp unidirectional (UD) preforms and their composites. On the whole, the overall originality of the concept used here is medium-high. Nevertheless, I would recommend publication of this article in Molecules on the condition a major revision of the manuscript will be carried out and the following points will be taken into consideration.
We are thankful to the reviewer for suggesting following corrections and recommending this article for the publication in the Molecules journal. We have worked and accommodated all quarries of the reviewer and updated the manuscript accordingly in order to increase the readability of the paper. Our responses to the comments of the reviewer has been marked mostly as blue color text track changes in the manuscript.
Detailed comments:
The abstract needs to be well written with future prospects of the work and describe in short, the concept of preparation of yarn non-crimp unidirectional preforms.
We thank the reviewer and based on the suggestion, the abstract has been updated.
There is a lack of important references related to the e.g. methods used for receiving presented material. Furthermore, the introduction should be worked out - so as to show the full state of knowledge on this topic.
Yarn based UD structure considering natural yarn is limited in the literature. We tried to make a systemic approach to define the current state of the problems and analyze the background of the novel methods that adopted in this study and the hypothesis was to develop the UD non-crimped jute yarn based preforms for composites. We have updated the introduction section and please kindly see the track changes on the introduction section of the manuscript.
More detailed results discussion should be provided. The discussion section appears to be a collection of data, however, the author’s self-opinion is of importance while drafting a results section of this type.
Kindly see the track changes in different colors and updated version of the discussion section.
The conclusion reflects an overall summary of the field with further extension and includes future prospective - I would suggest clarifying this section.
The conclusion section has been updated according to the reviewer’s suggestions.
The style and grammar leave much to be desired in many places, some parts of the text are difficult to understand.
We have checked the grammar thoroughly and please see the latest version to find the modification of sentences where it was needed
Reviewer 2 Report
The manuscript “Fabrication and mechanical performance of non-crimped unidirectional jute yarn preform based composites” mainly investigated and developed yarn-based non-crimped unidirectional preforms from jute yarns. This topic corresponds to the journal's scope, and this article content shows interesting results. Some comments are as follows:
- In the section 3.4.3., please describe the dimensions of the sample and the span of a double cantilever bending mode. Additionally, the symbol “Eo” was not observed in your article. Please revise it.
- In Table 2, the standard deviation is very high for the preform thickness, especially for UD20, UD25, and UD30. Please check it.
- In the article, the authors showed the data with standard deviation in Figs.7 and 8 and Tables 2 and 4. It should perform statistical analyses on data because some results may be sufficient or insufficient difference between samples in the experiment.
- In the section 4.4 for theoretical analysis, the authors describe “The results indicate that the composite tensile property from the experimental data…..(Page 13, Lines 411-413)”. However, the difference between theoretical and experimental data is still observed. Please show the error of the data and explain the possible reasons for this result or how to improve it.
Author Response
Overall Comments: The manuscript “Fabrication and mechanical performance of non-crimped unidirectional jute yarn preform based composites” mainly investigated and developed yarn-based non-crimped unidirectional preforms from jute yarns. This topic corresponds to the journal's scope, and this article content shows interesting results. Some comments are as follows.
We are thankful to the reviewer for the consideration of our work and valuable comments to improve the work further. Our responses to the reviewer’s comments have been addressed mostly in yellow color text in the revised manuscript.
Comments:
- In the section 3.4.3., please describe the dimensions of the sample and the span of a double cantilever bending mode. Additionally, the symbol “Eo” was not observed in your article. Please revise it.
We thankful to the reviewer to identify this. This information has been updated in the manuscript at section 3.4.2 and 3.4.3 and marked the changed words in yellow color.
- In Table 2, the standard deviation is very high for the preform thickness, especially for UD20, UD25, and UD30. Please check it.
We regret for the mistake. The correct information produced from the calculated data is now provided in Table 2. This is marked in yellow color in the Table 2.
- In the article, the authors showed the data with standard deviation in Figs.7 and 8 and Tables 2 and 4. It should perform statistical analyses on data because some results may be sufficient or insufficient difference between samples in the experiment.
We acknowledge the reviewer for pointing out this concern regarding scattering effects of different specimen types. We know that single jute fibre tensile properties can be varied due to the irregularity in the fibre fineness. However, in composite level, the variation was reduced. We therefore, conducted a general statistical analysis and observed the co-efficient of variation of the specimen type is less than 10%. As fact of that, average value is reported in the study. We described this in results and discussion section 4.1, marked in the yellow color.
- In the section 4.4 for theoretical analysis, the authors describe “The results indicate that the composite tensile property from the experimental data .(Page 13, Lines 411-413)”. However, the difference between theoretical and experimental data is still observed. Please show the error of the data and explain the possible reasons for this result or how to improve it.
We thank the reviewer for finding the difference in the theoretical and experimental tensile properties of composites. It is expected that theoretical values should be higher than the experimental values. The lower experimental values are related with the uniformity of the fibre. Natural fibres are not uniform along the length of the yarn. Composite fails in the weakest zone present in the fibre is mainly responsible for the reduction in experimental values. Whereas in theoretical value, we assume that all the way fibres in the yarn have uniform quality and fails uniformly. Please check the updated discussion in section 4.4, marked in yellow color.
Reviewer 3 Report
Comments:
The submitted manuscript is consistent with the scope and readership of the journal. However, authors need to consider the following comments before publication:
1. Line 46, "jute is the second most abundant natural 46 fibre after cotton" not sure if jute really is listed 2nd, rather than flax.
2. Line 92, "To avoid this issue, glass filament bundle with same diameter and lineardensity can be used in..." it would be difficult to find such glass filament enjoying same density (volumetric or linear) as natural fibre.
3. For Materials part, the physical an mechanical properties of E-glass fibre used for weft yarns are missing, it is suggested to include them in Table 1.
4. Figure 1, it is hard to read printed words labelled on these photos, these images can move to Supplementary.
5. Figure 2 can be combined with Figure 3, as they both basically describe the preform preparation details.
6. Line 168-170, was the cured composite difficult to demount from the compression machine? It is possible that extra curable resin acts like a glue to fix the two plates of the hot press machine.
7. Line 231-233, not just because a much closer weft yarn spacing, but also higher density of glass fibre itself compared to jute fibre.
8. Line 264, to be exact, these individual fibres are elementary fibre s within a fibre filament [Cellulose (2019) 26:4693]
9. Figure 7 b, c and Figure 8 b, c provide same information as that in Table 4, so they can be deleted. Furthermore, high-resolution curves, bar charts and images are needed.
10. Line 379, Did author measured interfacial shear strength parameter t, in order to calculate critical fibre length according to Equation 3? The detailed experimental procedures for t measurement should be included in Experimental
11. Thermo-mechanical properties of the polyester matrix plays a major role in determining its correspondent composite performance. Therefore, in tensile, flexural, and DMA curves (Fig., 7, 8, 10, 100), relevant results from neat polyester have to be added for comparison.
12. Funding and acknowlegement information are missing
13. In Results and Discussion section, there are redundant information known as background knowledge appearing many times, which should be removed. And more concise language expression is expected.
Author Response
We are grateful to the reviewer for pointing out important issues in the paper which needs to address in the revised manuscript. Here we went through your individual quarries and changed in the manuscript accordingly. Our responses to the reviewer’s comments can be seen in green color text in the revised manuscript
Overall Comments: The submitted manuscript is consistent with the scope and readership of the journal. However, authors need to consider the following comments before publication.
We are grateful to the reviewer for pointing out important issues in the paper which needs to address in the revised manuscript. Here we went through your individual quarries and changed in the manuscript accordingly. Our responses to the reviewer’s comments can be seen in green color text in the revised manuscript.
Comments:
- Line 46, "jute is the second most abundant natural fibre after cotton" not sure if jute really is listed 2nd, rather than flax.
We appreciate the reviewer for this comment. We have slightly changed the wording to avoid any confusion regarding jute and flax fibres.
- Line 92, "To avoid this issue, glass filament bundle with same diameter and linear density can be used in..." it would be difficult to find such glass filament enjoying same density (volumetric or linear) as natural fibre.
We are thankful to the reviewer for finding out these differences between jute and glass fibres. We agree with reviewer that same diameter and linear density of glass filaments would be difficult to find. We therefore used traditional E-glass flat filaments (roving) in the weft direction where number of filaments were 200 of having similar diameter and linear densities (all filaments). We also modify the sentences in the introduction part can be found in line no. (100 – 104) , marked in green color.
For Materials part, the physical and mechanical properties of E-glass fibre used for weft yarns are missing, it is suggested to include them in Table 1.
We appreciate the reviewer for this concern. We have now included the information of E-glass fibre in Table 1, marked in green color.
Figure 1, it is hard to read printed words labelled on these photos, these images can move to Supplementary.
We acknowledge the reviewer for finding the issue related with digital images of jute yarns. The figure has been transferred in the supplementary files and in the footnote we clearly explained which spool was used for particular yarn types.
Figure 2 can be combined with Figure 3, as they both basically describe the preform preparation details.
According to the reviewer suggestion figure 2 and 3 have been merged together in the Figure 1 of the revised manuscript.
Line 168-170, was the cured composite difficult to demount from the compression machine? It is possible that extra curable resin acts like a glue to fix the two plates of the hot press machine.
We used Teflon sheet at the top and bottom side of the impregnated preforms so that it can be easily demolded. Moreover. we used a vacuum plastic bag to collect the extra flashed resin. So, there was no possibility of gluing of resin with metal parts of compression machine.
Line 231-233, not just because a much closer weft yarn spacing, but also higher density of glass fibre itself compared to jute fibre.
We agree with the reviewer and have included this reason in the discussion, please check line no. (241-243) in the revised manuscript.
Line 264, to be exact, these individual fibres are elementary fibre s within a fibre filament [Cellulose (2019) 26:4693]
We agree with the reviewer that individual fibres are regarded as elementary fibres. But they are not considered as filaments. In the case of plant fibres, the elementary fibres are cemented together which generally identify as technical fibres. Depending on the linear density of plant yarn the fibres individualization may vary. As a fact of that composite properties also significantly affected.
Figure 7 b, c and Figure 8 b, c provide same information as that in Table 4, so they can be deleted. Furthermore, high-resolution curves, bar charts and images are needed.
As the information of figure 7 and 8 (b,c) and Table 4 are same therefore based on the recommendation of the reviewer, Table 4 has been excluded from the manuscript and changed the tracked information accordingly.
Line 379, Did author measured interfacial shear strength parameter t, in order to calculate critical fibre length according to Equation 3? The detailed experimental procedures for t measurement should be included in Experimental
We are aware of the missing information of required reference suited for interfacial shear strength. We used our previously reported IFSS considered for IFSS of jute and epoxy[1]. We assumed jute has similar IFSS with polyester like flax fibre showed in the comparison of IFSS with different matrices (epoxy, polyester and bio-epoxy) [2]. As flax and jute has almost similar chemical composition and mechanical behaviour in composites therefore, we used the similar values of IFSS from the previously published article in our study.
Thermo-mechanical properties of the polyester matrix plays a major role in determining its correspondent composite performance. Therefore, in tensile, flexural, and DMA curves (Fig., 7, 8, 10, 100), relevant results from neat polyester have to be added for comparison.
In this work, our focus was to develop non-crimp UD jute preforms for composites and compare different linear densities of jute yarns having different twist levels for the analysis of mechanical and thermo-mechanical response of jute/polyester composites. All composites were manufactured using the same polyester matrix. This study clearly demonstrated that yarn linear densities and twist level differences affect tensile, flexural and storage modulus along with damping properties significantly. Therefore, we didn’t compare the neat polyester response in this regard. The time preparation and characterization of polyester matrix samples will be time consuming. Considering the shortest available time for the revision and the above description in the picture, it will be appreciated if the reviewer consider this limitation and allow the current comparison.
Funding and acknowledgement information are missing
We thank the reviewer for this comment. We have now included this information in the manuscript.
In Results and Discussion section, there are redundant information known as background knowledge appearing many times, which should be removed. And more concise language expression is expected.
We highly acknowledge the reviewer for this comment. We have now edited and updated this section.
References
[1] F. Sarker, N. Karim, S. Afroj, V. Koncherry, K. S. Novoselov, and P. Potluri, “High-Performance Graphene-Based Natural Fiber Composites,” ACS Appl. Mater. Interfaces, vol. 10, no. 40, pp. 34502–34512, 2018, doi: 10.1021/acsami.8b13018.
[2] F. Sarkar, M. Akonda, and D. U. Shah, “Mechanical properties of flax tape-reinforced thermoset composites,” Materials (Basel)., vol. 13, no. 23, pp. 1–16, 2020, doi: 10.3390/ma13235485.
Round 2
Reviewer 1 Report
I would like to support this revised paper (Manuscript Number molecules-1439462) for publication in Molecules. All suggested changes were made (or discussed/clarified) by the authors.
The results are informative, and the discussion is clear. Furthermore, this review article includes a balanced, comprehensive, and critical view of the research area.
To summarize, I think that this paper can be published as-is.
Reviewer 3 Report
The authors have corrected and replied my concerns in the revised manuscript. Overall, I agree to publish the improved work in Molecules.